# *Bacillus subtilis* Provides Long-Term Protection in a Murine Model of Allergic Lung Disease by Influencing Bacterial Composition

**Rosalinda Monroy Del Toro [1], Ryan Incrocci [1], Olivia Negris [2], Shaina McGrath [2]** and **Julie A. Swartzendruber [1,*]**

1  Department of Microbiology and Immunology, College of Graduate Studies, Midwestern University, Downers Grove, IL 60515, USA
2  Biomedical Sciences Program, College of Graduate Studies, Midwestern University, Downers Grove, IL 60515, USA
*  Correspondence: jswart@midwestern.edu

**Abstract:** Probiotics are an attractive target for reducing the incidence of allergic disease. *Bacillus subtilis* is a gut-associated probiotic bacteria that can suppress allergic lung disease; however, it is not clear for how long this protection lasts. We exposed C57Bl/6 mice to *B. subtilis* via oral gavage and challenged them with intranasal house-dust mite for up to 8 weeks. We found that *B. subtilis* treatment was able to provide protection from eosinophil infiltration of the airways for 3 weeks. This loss of protection correlated with an increase in the eosinophil chemoattractant CCL24. Additionally, we demonstrate that *B. subtilis* treatment altered the bacterial composition by increasing the phylum *Bacteroidetes* and *Verrucomicorbiota.* The phylum *Verrucomicorbiota* was reduced in *B. subtilis*-treated mice at 8 weeks when protection was lost. These results support *B. subtilis* as a prophylactic for preventing the production of allergic lung disease and highlights that protection can last up to 3 weeks. This work also expands our understanding of how *B. subtilis* mediates protection and that in addition to modifying the immune system it is also altering the host microbiota.

**Keywords:** probiotic; allergy; house-dust mite; eosinophils; host microbiota





## 1. Introduction

The number of people with allergies has been on the rise over the last half-century, and currently 20% of the population in developed countries has allergies [1]. The rise in allergic disease correlates with an increasingly sterile living environment and reduced exposure to bacteria, contributing to inappropriate responses by the immune system to non-infectious or innocuous antigens [2]. The rise in allergic disease and reduced early-life exposure to microbes is reflected in the changing bacterial composition of infants. Specifically, children born via caesarean section have reduced bacterial diversity and reduced levels of Lactobacillus and Bifidobacterium, and therefore increased risk of developing allergic disease [3,4]. Probiotic intervention is being pursued as a method to restore this bacterial dysbiosis. Both Lactobacillus and Bifidobacterium haven been shown to reduce the development of allergic disease in mice when administered prophylactically [5,6]. The mechanism of Lactobacillus protection is strain-specific and includes induction of T-regulatory cells, as well as modulation of dendritic cell function [7–9]. Although Lactobacillus is generally safe, it requires frequent dosing for efficacy, which poses a risk for immunocompromised patients and can lead to bacteremia [10].

We and others have identified that *Bacillus subtilis*, a Gram-positive bacterium commonly found in soil and the gastrointestinal tract, can prevent allergic lung inflammation [11,12]. *B. subtilis* has demonstrated efficacy in protecting from gastrointestinal inflammation, sepsis, and allergic lung disease with infrequent dosing in acute models of

disease [13,14]. *B. subtilis* has been demonstrated to reduce pathogenic bacteria via quorum sensing with implications for reducing infections [15]. In addition, we know that *B. subtilis* production of exopolysaccharide (EPS) is critical for its immune suppression [13,16]. Previous work has established that *B. subtilis*-derived EPS induces an anti-inflammatory response by suppressing dendritic-cell function and inducing anti-inflammatory macrophages, as well as via NF-kb and STAT6 pathways [11,12,16].

The process of allergic sensitization is initiated by antigen-presenting cells (APC), such as dendritic cells (DC), which capture and internalize allergens and are essential for HDM-induced type-2 inflammation [17]. Allergen presentation by DCs drives the proliferation of allergen-specific Th2 cells and activation of B cells [18,19]. Eosinophils are recruited to the airway following Th2-produced cytokines IL-4, IL-5, and IL-13 stimulating a variety of cells to produce the eosinophil chemoattractants CCL11 and CCL24 [20,21]. Allergen-specific IgE is produced by B cells after an encounter with IL-4-producing Th2 cells and is dependent on the STAT6 pathway [19].

Our previous work identified that *B. subtilis* treatment can protect mice from HDM-induced eosinophilia, but we did not determine whether *B. subtilis* influenced eosinophil chemokines or allergen-specific immunoglobulin production. Additionally, we do not know how long *B. subtilis*-mediated protection lasts. To determine how long *B. subtilis* can protect mice from an allergic response, we utilized repeated HDM challenges and found that *B. subtilis* protects from HDM-induced eosinophilia for 3 weeks. We further determined *B. subtilis* reduced levels of CCL24, with no impact on immunoglobulin production. Interestingly, we found that *B. subtilis* exposure altered the composition of bacteria during periods of protection compared to no-treatment mice. These results support the long-term protective effects of *B. subtilis* exposure, further identify the effects of *B. subtilis* on the immune response, and point to additional impacts of *B. subtilis* exposure on the microbiota.

## 2. Materials and Methods

### 2.1. Mice

Male and female C57Bl/6 mice were purchased from Jackson Laboratory (Bar Harbor, ME, USA) and bred in house at Midwestern University, Downers Grove, IL, in a specific pathogen-free facility. All animal protocols were approved by the Midwestern University Institutional Animal Care and Use Committee.

### 2.2. Allergic Lung Disease Model

C57Bl/6 mice, age 6–10 weeks old, were treated through oral gavage with 10ˆ9 wild-type *B. subtilis* spores, 10ˆ9 *epsH* spores unable to produce exopolysaccharide (EPS), or sterile PBS as a negative control. *B. subtilis* and *epsH* spores were generated via exhaustion, as previously described [22]. After 24 h, mice were sensitized intranasally with 100 μg of house-dust mites (HDM, *D. pteronyssinus* XPB82S3A2.5, Stallergenes Greer, Lenoir, NC, USA) in 100 μL of sterile PBS. After 5 days of initial treatment, mice were once again treated orally with *B. subtilis* spores as described above. On days 6–9, mice that were in the "week 1" group were sensitized intranasally with 25 μg HDM in 100 μL sterile PBS. These mice were euthanized on day 12. Bronchial alveolar lavage (BAL) was collected by flushing lungs with 0.8 mL BAL fluid (10% FCS, 1 mM EDTA, 1X PBS). Blood was collected through cardiac puncture and spun down at 8000 rpm for 8 min, and serum was collected and frozen for ELISA assays. Lungs were excised and fixed in formalin for 24 h. BALF was counted using a hemocytometer to determine total cells/mL, cytospun onto slides, and assessed for percent eosinophils, neutrophils, lymphocytes, and macrophages using Diff-Quick staining and counting 100 cells per slide (Dade Behring, Deerfield, IL, USA). The total cells/mL of each cell type was determined by calculating the number of cells/mL based on the percent of each cell type. On days 13–16, mice that were in the "week 2" group were sensitized intranasally with 25 μg HDM in 100 μL sterile PBS. On day 19, these mice were euthanized, and tissues were collected as described above. Intranasal sensitizations

were repeated for groups "week 3" through "week 8," mice were euthanized, and tissues were collected.

### 2.3. ELISA

*CCL24:* 96-well plates were coated with 100 μL of biotinylated anti-mouse CCL24 in carbonate buffer at 4 °C overnight (Peprotech, Cranbury, NJ, USA). The plates were washed three times with PBST buffer (1xPBS, 0.05% Tween 20) and blocked with 3% BSA in PBS for 2 h at room temperature. The plates were washed again and BALF was added to the wells along with antibody standard recombinant murine CCL24 (Peprotech). After three washes, the plates were treated with secondary antibody Biotin anti-murine CCL24 (Peprotech) and incubated at room temperature for 1 h. The plates were washed and stained with HRP-avidin (Biolegend, San Diego, CA, USA) in the dark at room temperature for 1 h. After washing the plates, TMB substrate solution (Biolegend) was added to the wells, and once there was a color change the plates were measured on a MultiSkan™ plate reader (Thermo Fisher, Waltham, MA, USA) at OD 405 nm.

*HDM-specific IgE and IgG1:* 96-well plates were coated with 100 μL of HDM (5 μg/μL) in carbonate buffer at 4 °C overnight. The plates were washed three times with PBST buffer (1xPBS, 0.05% Tween 20) and blocked with 3% BSA in PBS for 2 h at room temperature. The plates were washed again and serum was added to the wells. After subsequent washes, the plates were treated with secondary antibody Biotin anti-mouse IgE or IgG1 and incubated at room temperature for 1 h. The plates were washed and stained with HRP-avidin (Biolegend) in the dark at room temperature for 1 h. After washing the plates, TMB substrate solution (Biolegend) was added to the wells and once there was a color change, the plates were measured on a MultiSkan™ plate reader (Thermo Fisher, Waltham, MA, USA) at OD 405 nm.

*Total IgE and IgG1:* 96-well plates were coated with 100 μL of rat anti-mouse IgE or anti-mouse IgG1 (BD Biosciences, San Jose, CA) in carbonate buffer at 4 °C overnight. The plates were washed three times with PBST buffer (1xPBS, 0.05% Tween 20) and blocked with 3% BSA in PBS for 2 h at room temperature. The plates were washed again and serum was added to the wells along with antibody standards (BD Biosciences). The plates were incubated at 4 °C overnight. The plates were washed and stained with HRP-avidin (Biolegend) in the dark at room temperature for 1 h. After washing the plates, TMB substrate solution (Biolegend) was added to the wells and once there was a color change, the plates were measured on a MultiSkan™ plate reader (Thermo Fisher, Waltham, MA, USA) at OD 405 nm.

### 2.4. DNA Extraction and 16S rRNA Amplicon Sequencing

Fecal samples were collected from mice during the allergic lung inflammation model on days −1 and 5 (before o.g. treatment) and the day of euthanization across all treatment groups into sterile tubes and stored at –80 °C. DNA extraction was performed on thawed fecal samples using the E.Z.N.A. Tissue DNA Kit (Omega BioTek, Norcross, GA, USA) according to the manufacturer's instructions. The extracted DNA was sent to Argonne National Laboratory for 16S rRNA amplicon sequencing using the Illumina MiSeq instrument. The sequences obtained were processed through Illumina-utils software to demultiplex, align, merge, and quality filter the sequence data. Taxonomy was assigned using the SILVA database version 138. Alpha- and beta-diversity was determined using QIIM31.9.1 and QIIME2 to generate PCoA ordinations.

### 2.5. Statistics

All analyses were carried out using GraphPad Prism Software (La Jolla, CA, USA).

## 3. Results

### 3.1. Bacillus Subtilis Protection from House-Dust Mite-Induced Eosinophilia Wanes after 3 Weeks

To determine how long *B. subtilis* can protect against eosinophilia, we challenged mice with HDM for up to 8 weeks after *B. subtilis* exposure, as shown in Figure 1A. We measured eosinophils in the bronchial alveolar fluid lavage (BALF) and found that at 4 weeks mice receiving *B. subtilis* were no longer protected from eosinophilia (Figure 1B,C). Interestingly, the protection from weeks 2 and 3 seemed to be independent of exopolysaccharide production, as demonstrated by the reduced eosinophils in mice treated with *epsH* spores that were unable to produce EPS (Figure 1B,C). These results expand on our previous knowledge that EPS is important in the *B. subtilis*-mediated protection from eosinophilia at week 1 [11]. The reduction in eosinophils was due to a shift to macrophages being the primary cell in the BALF, as shown in Figure 2A,D. Additionally, *B. subtilis* treatment did significantly alter neutrophil infiltration and had a moderate impact on lymphocyte recruitment (Figure 2B,C,E,F).

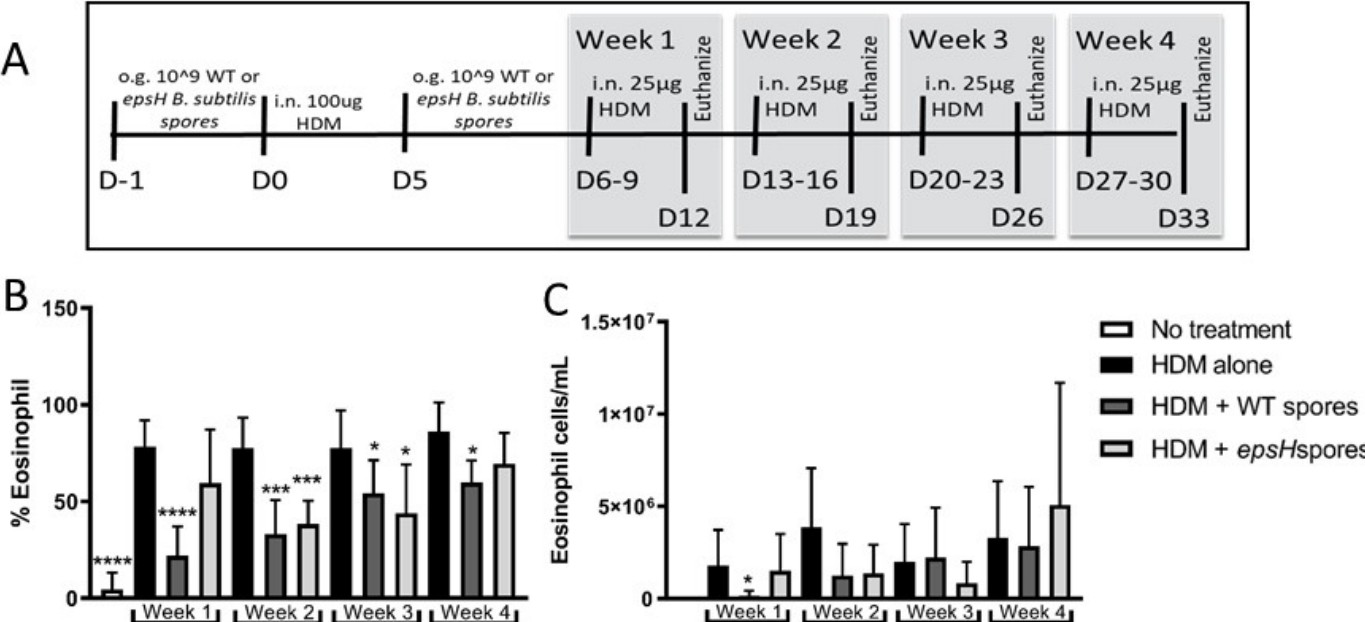

**Figure 1.** Eosinophil infiltrate in BALF following *B. subtilis* treatment. (**A**) Timeline for *B. subtilis* and HDM treatment in C57Bl/6 mice. (**B**,**C**) Quantification of eosinophils from BALF cytospun and stained with DiffQuick, as shown in (**A**). Data represent mean $\pm$ SEM representative ($n$ = 3–10). \* $p < 0.5$, \*\*\* $p < 0.005$, and \*\*\*\* $p < 0.001$ by Student $t$ test compared to HDM alone in the same week. HDM, house-dust mite; WT, wild-type; *epsH* spores, *B. subtilis* spores unable to produce exopolysaccharide.

### 3.2. CCL24 Production Is Suppressed by Bacillus subtilis Treatment for 3 Weeks

Eosinophil recruitment to the lung is driven in part by the production and presence of the chemokine CCL24. We measured the levels of CCL24 in the BALF each week for 4 weeks according to the *B. subtilis* exposure and HDM treatment outlined in Figure 1A. We found that CCL24 levels were significantly reduced for the first 3 weeks, but not after 4 weeks following *B. subtilis* exposure (Figure 3). Again, we found that suppressed CCL24 at week 3 was independent of EPS production, but EPS was involved in the CCL24 suppression at weeks 1 and 2 (Figure 3).

### 3.3. B. subtilis Treatment Does Not Influence Immunoglobulin Production

House-dust mite-induced eosinophilia is driven by the adaptive immune response and includes involvement of immunoglobulins IgG1 and IgE. To determine whether *B. subtilis*-mediated protection was influenced by a reduction in IgG1 or IgE production, we measured

serum levels of total and HDM-specific IgG1 and IgE. We found that *B. subtilis* treatment had no effect on HDM-specific IgG1 and IgE or total IgG1 and IgE levels (Figure 4A–D).

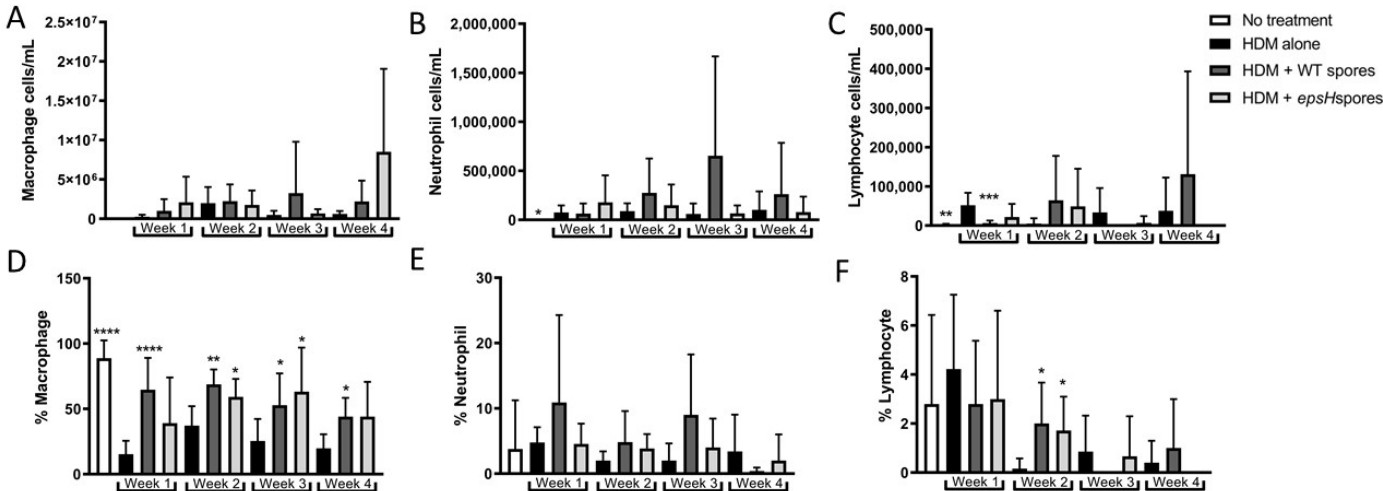

**Figure 2.** Cellular infiltrate of BALF following *B. subtilis* treatment. (**A–F**) Quantification of BALF cells from cytospin and stained with DiffQuick, as shown in timeline from Figure 1A. Data represent mean $\pm$ SEM representative ($n$ = 5–10). * $p < 0.5$, ** $p < 0.05$, *** $p < 0.005$, and **** $p < 0.001$ by Student $t$ test compared to HDM alone in the same week.

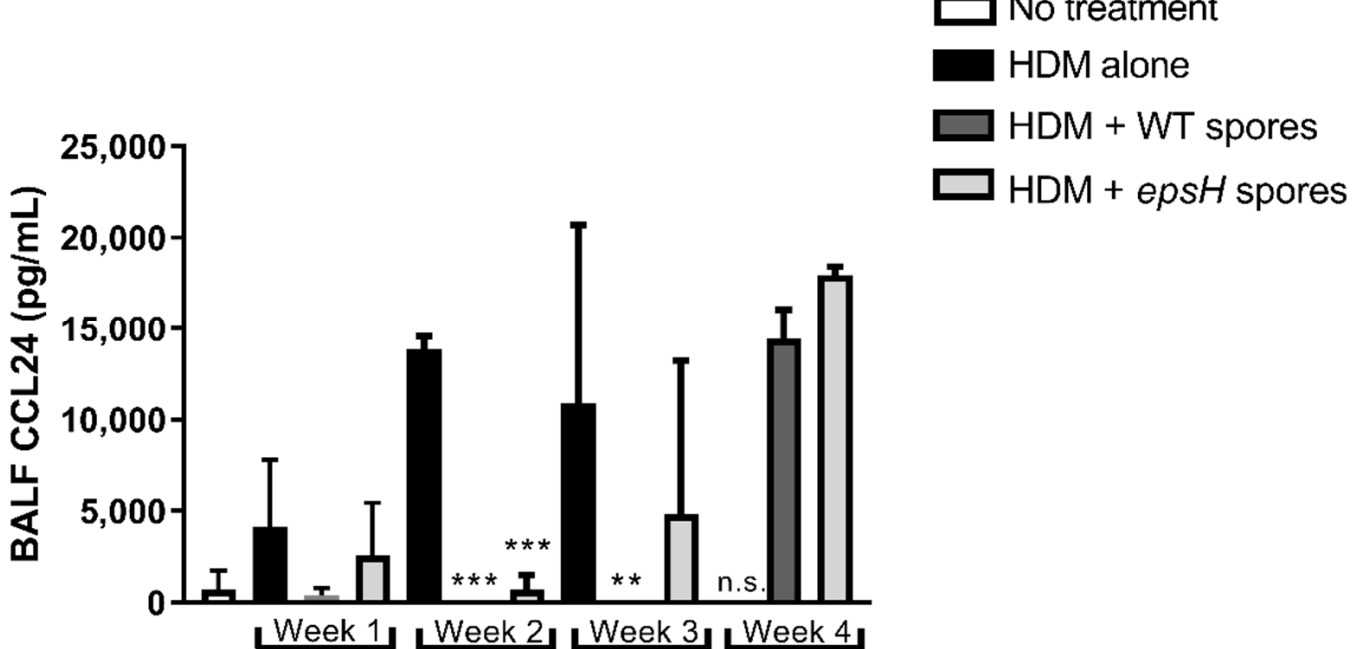

**Figure 3.** CCL24 in BALF following *B. subtilis* treatment. Quantification of CCL24 in BALF supernatant analyzed by ELISA. Data represent mean $\pm$ SEM representative ($n$ = 3–10). ** $p < 0.05$ and *** $p < 0.005$ by one-way ANOVA followed by Tukey's multiple-comparisons test was performed compared to HDM alone in the same week. n.s, no sample.

### 3.4. Bacillus Fecal Composition Is Similar across Mice Regardless of Treatment with B. subtilis Spores

To determine whether the waning protection was due to higher levels of *B. subtilis* following exposure that dwindled over time, we collected fecal pellets from mice over the course of 8 weeks to assess bacterial composition. We found that mice receiving no *B. subtilis* and HDM alone and mice receiving WT *B. subtilis* spores had very similar levels

of *Bacillus* strains identified by 16S rRNA amplicon sequencing (Figure 5A). Of note, mice treated with *epsH B. subtilis* spores had elevated levels of *Bacillus* during the first week following spore administration (Figure 5A). This elevated *Bacillus* seemed to be temporarily detrimental, as evidenced by the elevated eosinophilia in *epsH B. subtilis*-treated mice during week 1 and protection in weeks 2–3 (Figure 1B).

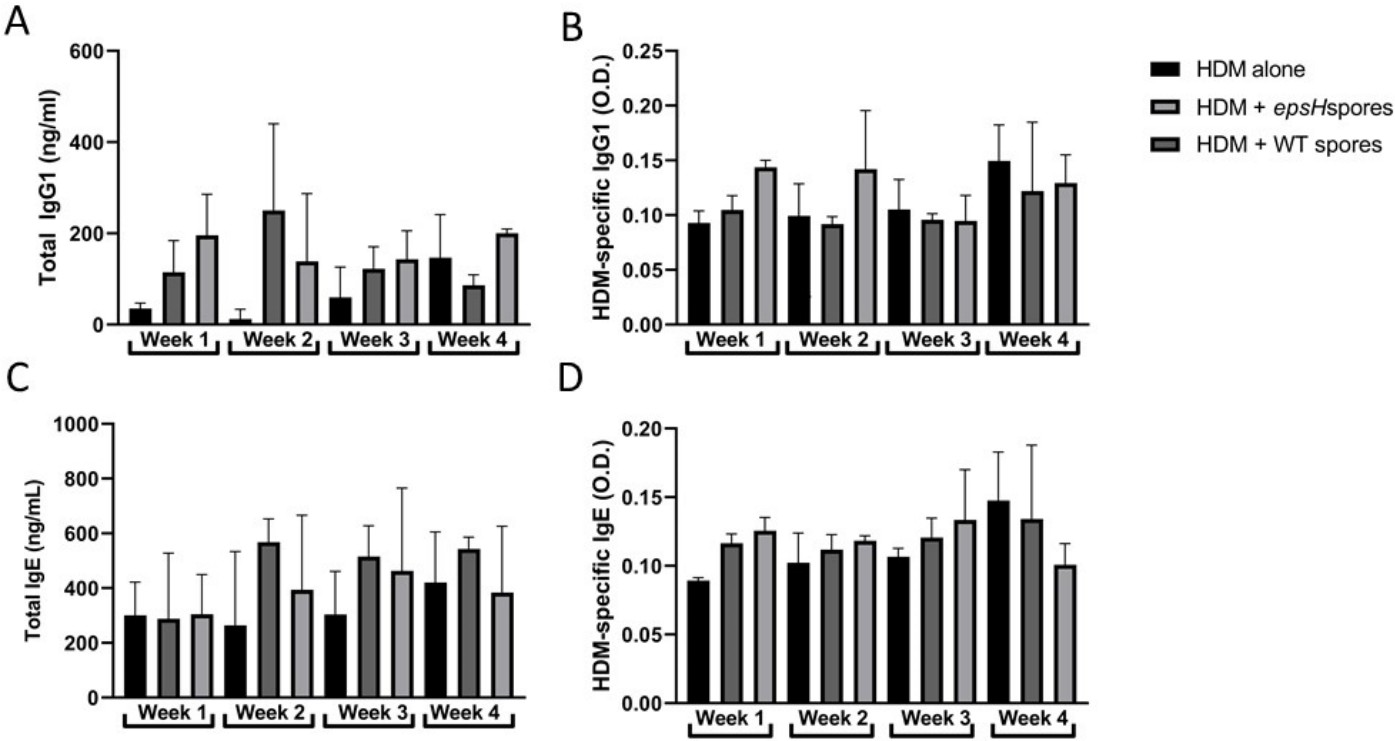

**Figure 4.** Immunoglobulin levels in serum following *B. subtilis* treatment. (**A**–**D**) Quantification of serum immunoglobulin levels by ELISA. Data represent mean $\pm$ SEM representative (*n* = 2–4). One-way ANOVA followed by Tukey's multiple-comparisons test was performed for each immunoglobulin and no statistical differences were found.

*3.5. Bacterial Composition Clustering Correlates with Protection from Eosinophilia*

This early difference induced between *epsH* and WT *B. subtilis* was also observed when comparing all bacterial compositions, as evidenced by all three treatment groups clustering separately early in treatment and *epsH* spores clustering more closely with HDM alone during the time of elevated eosinophilia. However, at day 5 *epsH* spore treatment changed its clustering to be more closely related to WT *B. subtilis* spores, correlating to a shift to protection from eosinophilia (Figure 5B). Additionally, we found that bacteria phylum was altered by *B. subtilis* treatment, specifically increased levels of the phylum Bacteroidetes, as well as elevated Verrucomicorbiota early in disease during *B. subtilis*-mediated protection (Figure 5C). In addition to elevated Bacteroidetes and Verrucomicorbiota in *B. subtilis* treatment, there was reduced representation by the phylum Firmicutes (Figure 5C).

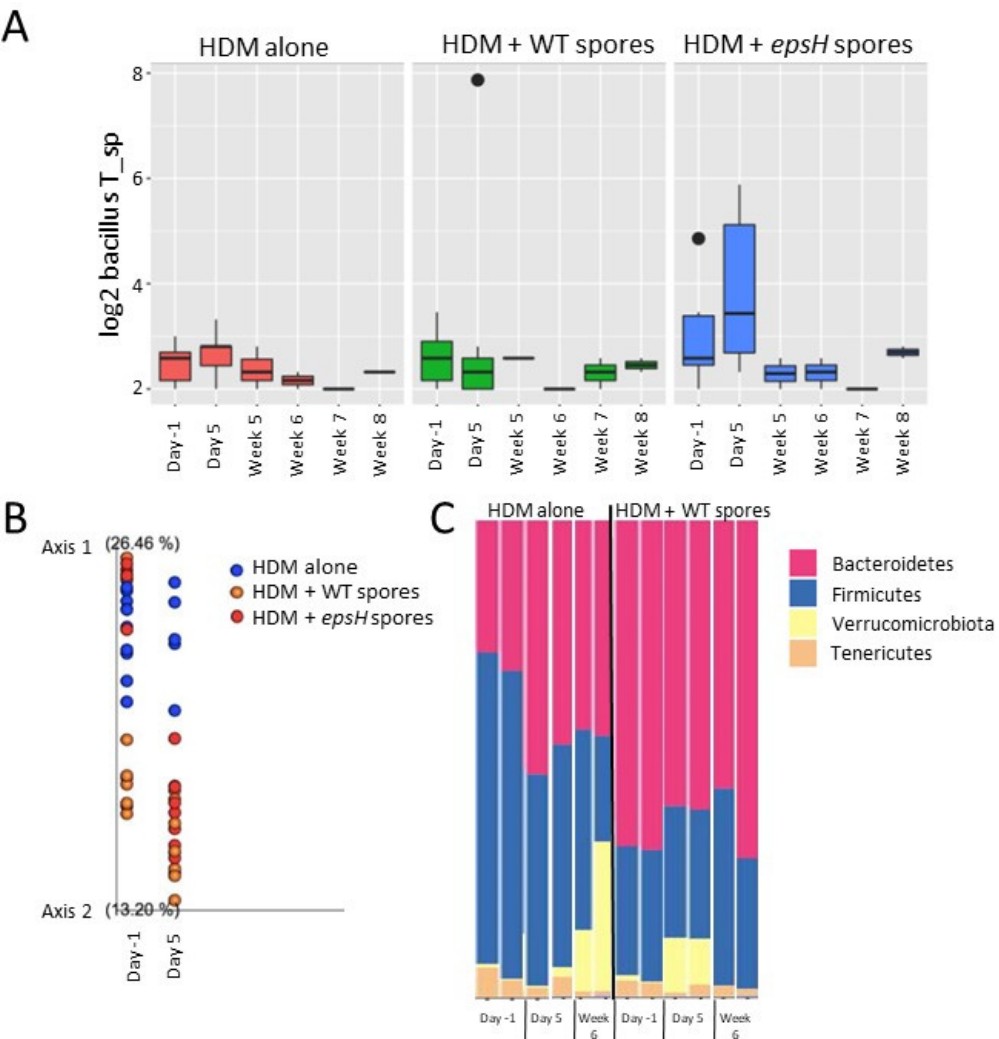

**Figure 5.** Bacterial composition following *B. subtilis* treatment. (**A**) Quantification of *Bacillus* strains in 16S rRNA amplicon sequencing from fecal pellets of mice treated according to the experimental design in Figure 1A. (**B**) PCoA plot between HDM alone, HDM + WT spores, and HDM + *epsH* spores on a forced axis for treatment. (**C**) Taxonomic bar plot indicating relative abundance of bacterial phylums.

## 4. Discussion

This work expands our understanding of the duration of protection provided by *B. subtilis* and offers further evidence for its appeal as a probiotic for preventing allergic disease. Unlike other probiotics that require early and frequent dosing, *B. subtilis* is effective at 6 weeks of age and last for up to 3 weeks. For example, many of the experiments investigating the efficacy of the probiotic lactobacillus utilize daily dosing [5]. *B. subtilis* provides a novel bacterium to prevent allergic disease with infrequent dosing required.

Current treatments for allergic disease focus on improving symptoms while rarely addressing the mechanism of disease [23]. Treatments that do address the mechanism, such as desensitization studies, involve high patient risk for anaphylactic reactions, are time consuming and expensive, and have only moderate efficacy [24]. We identified *B. subtilis* as a candidate for treating allergic disease and further characterized *B. subtilis*-derived EPS as a bacterial-derived molecule that elicits immunosuppression. Specifically, we found that exposure to *B. subtilis* WT spores resulted in reduced production of eosinophilia at week 1, whereas *epsH* spore treatment did not show a significant decrease in eosinophils at this timepoint. Additionally, we found that exposure to *B. subtilis* EPS-producing WT spores resulted in reduced production of the eosinophil chemokine CCL24 at week 1, whereas

EPS was not required for *B. subtilis* treatment to reduce CCL24 levels or eosinophilia at week 3. CCL24 is produced by many cell types in response to IL-4, including airway epithelial cells, airway smooth-muscle cells, vascular endothelial cells, macrophages, and eosinophils [25–27]. We suspect that the mechanism of *B. subtilis* WT spores and *epsH* spores reducing CCL24 at different type points is due to suppression of CCL24 production by different cell types.

There is growing evidence that the resident microbiota influences susceptibility to allergic disease [28]. Although we and others have identified the immunosuppressive effects of *B. subtilis* and have investigated the effects on immune-cell function and activation, no one had previously investigated the impact on the host microbiota. We found that treatment with *B. subtilis* spores influences the bacterial composition of treated mice. Interestingly we found minimal change in *Bacillus* strains observed by 16S rRNA amplicon sequencing, which is in line with previous reports showing that *B. subtilis* spore treatment does not establish elevated colonization, with CFUs/gm feces declining rapidly and reaching the limit of detection around 4 days post exposure [29]. Our work highlights the importance of host microbiota on allergic disease and sheds light on how probiotics can influence bacterial composition.

Although allergies are increasing in prevalence and contribute to reductions in quality of life, as well as significant health-care costs [30], we are pursuing a novel method for reducing the incidence of disease using an abundant, cheap, and safe probiotic, *Bacillus subtilis.* Future work is needed to determine whether *B. subtilis* can serve as a treatment option after the onset of disease or whether its effects are limited to prophylactic treatment. Additionally, previous work has identified that *B. subtilis*-derived EPS immune suppression is TLR4 dependent [13], and future studies are needed to determine the requirement for TLR4 in protection from allergic airway inflammation.

**Author Contributions:** Conceptualization, J.A.S.; methodology, R.M.D.T., R.I., O.N. and S.M.; data analysis, R.M.D.T., R.I., O.N., S.M. and J.A.S.; writing—original draft preparation, R.M.D.T. and J.A.S.; writing—review and editing, R.M.D.T., R.I. and J.A.S.; funding acquisition, J.A.S. All authors have read and agreed to the published version of the manuscript.

**Funding:** This research received no external funding but was supported by Midwestern University Start-Up Funds and the Midwestern University Core Facility, including the Core Facility Outsourcing Fund, all awarded to J.A.S.

**Institutional Review Board Statement:** The animal-study protocol was approved by the Institutional Animal Care and Use Committee of Midwestern University, Downers Grove, IL (protocol code #2754, approval date 27 November 2018).

**Informed Consent Statement:** Not applicable.

**Data Availability Statement:** Not applicable.

**Acknowledgments:** We would like to acknowledge and thank the Argonne Sequencing Team for performing the 16S rRNA amplicon sequencing and Nathaniel Hubert for analysis of the 16S rRNA sequencing data.

**Conflicts of Interest:** The authors declare no conflict of interest.

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
