# Peer review of "Bacillus subtilis Provides Long-Term Protection in a Murine Model of Allergic Lung Disease by Influencing Bacterial Composition"

_allergies, doi:10.3390/allergies3010001_

Round 1

Reviewer 1 Report

The manuscript by Monroy del Toro et al provides data on how Bacillus subtilis provides 3-week effect on some parameters realted to allergy (such as eosinophilia or IgE levels) in a murine model of HDM- induced allergy. The effect of this bacteria sounds promising and microbiota manipulation is recently being a matter of interest for the treatment of inflammatory diseases. However, there are some methodological issues that need to be clarified and improved along the text and the figures. 

- Line 82 and 84 and further (also in the figures where needed)- Please use the Greek symbol for "micro".

- Line 141- Please, write analyses (in plural). Also, indicate the statistical tests and significance level used for every statistical comparison.

- Figure 1 legend- Please indicate what HDM, WT and epsHspores mean.

- Figures 1 & 2- Could you please explain how you calculated cell numbers and % from cytospin stainings and include it in M&M sections?

- Figure 3- I am surprised that there are not significant differences in CCL24 levels at weeks 2 and 4. Please, include the significance stars (if any) and the test used for comparisons. Also, where is the bar for HDM treatment alone at week 4 as well as those missing from weeks 2 and 3? Please, revise this graph. Describe in the text when an experimental group is missing and state the reason for it.

- Figure 4- In relation to the previous, why are some data missing at week 1 in this figure? Please, give a reason for this is the text. 

- Lines 194-195- Why do you state that the levels of IgG1 total and HDM-specific and IgE are elevated in HDM treatment? There is not statistical significance according to your graphs (Figure 4A-D).

- Lines 196-198- Please, indicate the week when this happens. It is not the case for every week. I guess you refer to week 4. I would not say that total IgE levels are different (or tend to be different) at all at week 4 between HDM alone and HDM+WT spores. Please, again revise your numbers, statistical results and graphs. Why do you use student t test for just n=2-4?

Author Response

The manuscript by Monroy del Toro et al provides data on how Bacillus subtilis provides 3-week effect on some parameters realted to allergy (such as eosinophilia or IgE levels) in a murine model of HDM- induced allergy. The effect of this bacteria sounds promising and microbiota manipulation is recently being a matter of interest for the treatment of inflammatory diseases. However, there are some methodological issues that need to be clarified and improved along the text and the figures. 

- Line 82 and 84 and further (also in the figures where needed)- Please use the Greek symbol for "micro".

Response – The Greek symbol for micro has been used in lines 82 and 84, as well as where appropriate in the rest of the document.

- Line 141- Please, write analyses (in plural). Also, indicate the statistical tests and significance level used for every statistical comparison.

Response – We have changed analysis was to analyses were, as requested.  Due to the statistical tests differing for different experiments we have provided the specific statistical tests and significance level used for statistical comparison in each figure legend or method description.

- Figure 1 legend- Please indicate what HDM, WT and epsHspores mean.

Response – The abbreviations have been explained in the figure legend.

- Figures 1 & 2- Could you please explain how you calculated cell numbers and % from cytospin stainings and include it in M&M sections?

Response – We have included more information in the Allergic Lung Disease Model section (lines 95-99) to include that the total cells per sample were determined using the hemacytometer and that the percent of each cell type was determined by counting 100 cells/slide and calculating the total number of cell type based on percent of total cells per sample.

- Figure 3- I am surprised that there are not significant differences in CCL24 levels at weeks 2 and 4. Please, include the significance stars (if any) and the test used for comparisons. Also, where is the bar for HDM treatment alone at week 4 as well as those missing from weeks 2 and 3? Please, revise this graph. Describe in the text when an experimental group is missing and state the reason for it.

Response - We have added the statistical analysis description and p values to Figure 3.  We performed a one-way ANOVA followed by Tukey’s multiple comparisons test.  We found that treatment with HDM + WT spores and epsH spores resulted in significantly less CCL24 in the BALF at week 2, while only HDM + WT spores significantly reduced CCL24 at week 3.  Week 4 levels of CCL24 in both HDM + WT and epsH spores were elevated to levels of HDM alone from weeks 2 and 3. The HDM treatment alone at week 3 has no sample, this has been added to the figure. 

- Figure 4- In relation to the previous, why are some data missing at week 1 in this figure? Please, give a reason for this is the text. 

Response – We apologize for this error and have replaced Figure 4 with data including week 1.

- Lines 194-195- Why do you state that the levels of IgG1 total and HDM-specific and IgE are elevated in HDM treatment? There is not statistical significance according to your graphs (Figure 4A-D).

Response – We have removed this statement and the description now states there is no effect of B. subtilis on immunoglobulin production.

- Lines 196-198- Please, indicate the week when this happens. It is not the case for every week. I guess you refer to week 4. I would not say that total IgE levels are different (or tend to be different) at all at week 4 between HDM alone and HDM+WT spores. Please, again revise your numbers, statistical results and graphs. Why do you use student t test for just n=2-4?

Response – As mentioned above, we have revised this figure and the results section of the figure to state that B. subtilis does not alter immunoglobulin production.  We have corrected the statistical description to state that a one-way ANOVA was performed followed by Tukey’s multiple comparison test and found no statistical difference for each immunoglobulin.

Reviewer 2 Report

In this paper, the authors explore how long B. subtilis-mediated protection will last, whether B. subtilis influences eosinophil chemokines and allergen-specific immunoglobulin production. The results are of interest to readers and have new insights. I have some comments listed below.

                     1. Page 4, Figure 1

1)      Bacillus subtilis protection from HDM-induced eosinophilia lasted for up to 4 weeks regarding eosinophil percentage in BALF but only for 1 week regarding eosinophil number. How do the authors explain this difference? Which one play a more important role in HDM-induced allergic lung disease, eosinophil percentage or absolute number?

2)      Besides BALF, did the authors measure the eosinophil levels in other organs, like spleen?

3)      The protection from weeks 2 and 3 seemed to be independent of EPS. What would the possible mechanism be? 

2. Page 5, Figure

 B. subtilis treatment decreased lymphocytes infiltration in BALF, did the author further determine what lymphocyte subsets would be affected? How about Th2 and B cells?

3. Page 5, Figure 3

1)      It was described in the Method that CCL24 levels were measured with serum while in the Figure 3 CCL24 levels were measured with BALF?

2)      Was there any correlation between BALF CCL24 and serum CCL24 levels?

3)      There was no description about how the statistical analysis was made regarding Figure 3. What were the P values?

Author Response

In this paper, the authors explore how long B. subtilis-mediated protection will last, whether B. subtilis influences eosinophil chemokines and allergen-specific immunoglobulin production. The results are of interest to readers and have new insights. I have some comments listed below.

  1. Page 4, Figure 1
  • Bacillus subtilis protection from HDM-induced eosinophilia lasted for up to 4 weeks regarding eosinophil percentage in BALF but only for 1 week regarding eosinophil number. How do the authors explain this difference? Which one play a more important role in HDM-induced allergic lung disease, eosinophil percentage or absolute number?

Response – While we don’t maintain statistically significant protection for as many weeks in the cells/mL graph, we believe the reduced percent eosinophils is pathologically significant.  There is evidence from other groups that B. subtilis activates anti-inflammatory macrophages and we are pursuing experiments to determine if B. subtilis treatment results in an accumulation of anti-inflammatory macrophage. This may contribute to an influx of cells at later time points that alters the total number of eosinophils.

  • Besides BALF, did the authors measure the eosinophil levels in other organs, like spleen?

Response – We did not measure eosinophils in any other organs.  The allergic lung inflammation model has been well characterized with eosinophils infiltrating through the blood to the lung.

  • The protection from weeks 2 and 3 seemed to be independent of EPS. What would the possible mechanism be? 

Response – We are not sure what the mechanism is for EPS independent protection at weeks 2 and 3.  There is previous literature that has found a role for flagella in B. subtilis protection from Citrobacter rodentium-associated enteric disease.

  1. Page 5, Figure
  2. subtilis treatment decreased lymphocytes infiltration in BALF, did the author further determine what lymphocyte subsets would be affected? How about Th2 and B cells?

      Response – We have not quantified the lymphocyte subsets including Th2 or B cells.  We agree this is an interesting question and have plans to determine the impact of B. subtilis on lymphocyte subsets in future experiments.

  1. Page 5, Figure 3

1)      It was described in the Method that CCL24 levels were measured with serum while in the Figure 3 CCL24 levels were measured with BALF?

Response – We have corrected the methods section to correctly describe that CCL24 was measured in the BALF.

2)      Was there any correlation between BALF CCL24 and serum CCL24 levels?

      Response – We have not measured serum CCL24 levels.  Our understanding is that CCL24 is produced locally in the airways by macrophages and alveolar type II cells and would therefore not be found at high levels in the serum.

3)      There was no description about how the statistical analysis was made regarding Figure 3. What were the P values?

Response – We have added the statistical analysis description and p values to Figure 3.  We performed a one-way ANOVA followed by Tukey’s multiple comparisons test.  We found that treatment with HDM + WT spores and epsH spores resulted in significantly less CCL24 in the BALF at week 2, while only HDM + WT spores significantly reduced CCL24 at week 3.  Week 4 levels of CCL24 in both HDM + WT and epsH spores were elevated to levels of HDM alone from weeks 2 and 3.

Round 2

Reviewer 1 Report

The authors have addressed my major concerns and corrected methodological issues along their manuscript.

Reviewer 2 Report

The authors have adequately addressed my questions. I have no further queries.